# Evaluation and Prediction of Landslide Susceptibility in Yichang Section of Yangtze River Basin Based on Integrated Deep Learning Algorithm

**Lili Chang [1], Rui Zhang [2,*] and Chunsheng Wang [1]**

[1] Institute of Geophysics and Geomatics, China University of Geosciences, Wuhan 430079, China; lilichang@cug.edu.cn (L.C.); atccll@cug.edu.cn (C.W.)

[2] Land Satellite Remote Sensing Application Center, Ministry of Natural Resources of the People's Republic of China, Beijing 100048, China

\* Correspondence: zhangrui@radi.ac.cn

**Abstract:** Landslide susceptibility evaluation (LSE) refers to the probability of landslide occurrence in a region under a specific geological environment and trigger conditions, which is crucial to preventing and controlling landslide risk. The mainstream of the Yangtze River in Yichang City belongs to the largest basin in the Three Gorges Reservoir area and is prone to landslides. Affected by global climate change, seismic activity, and accelerated urbanization, geological disasters such as landslide collapses and debris flows in the study area have increased significantly. Therefore, it is urgent to carry out the LSE in the Yichang section of the Yangtze River Basin. The main results are as follows: (1) Based on historical landslide catalog, geological data, geographic data, hydrological data, remote sensing data, and other multi-source spatial-temporal big data, we construct the LSE index system; (2) In this paper, unsupervised Deep Embedding Clustering (DEC) algorithm and deep integration network (Capsule Neural Network based on SENet: SE-CapNet) are used for the first time to participate in non-landslide sample selection, and LSE in the study area and the accuracy of the algorithm is 96.29; (3) Based on the constructed sensitivity model and rainfall forecast data, the main driving mechanisms of landslides in the Yangtze River Basin were revealed. In this paper, the study area's mid-long term LSE prediction and trend analysis are carried out. (4) The complete results show that the method has good performance and high precision, providing a reference for subsequent LSE, landslide susceptibility prediction (LSP), and change rule research, and providing a scientific basis for landslide disaster prevention.

**Keywords:** Yichang section of the Yangtze River Basin; non-landslide sample selection; SE-CapNet; landslide susceptibility evaluation; landslide susceptibility prediction

## 1. Introduction

In recent years, affected by global climate change, seismic activities, and accelerated urbanization, geological disasters such as landslides, collapses, and debris flows have increased significantly [1]. As a common geological disaster, landslides cause severe economic losses and unfortunate casualties and seriously block transport lines and waterways [2,3]. Especially in recent years, accelerated global change and rapid urbanization and industrialization have increased the likelihood of landslides, leading to more casualties and property losses [4–6]. The mountainous and hilly landforms in the Yichang section of the Yangtze River Basin are widely distributed. The geological environment is very fragile, and there are many geological disasters such as landslides, collapses, and debris flow [7–9]. Especially along the Three Gorges Reservoir area, the development of geological disasters is more intense, which poses a threat to urban construction and residents' production and life [10]. In recent years, accelerated global change, urbanization and industrialization have increased the likelihood of landslides, resulting in more casualties and property damage [11].

Landslide susceptibility evaluation (LSE) is the basis for landslide disaster risk assessment and prediction prevention, which can help relevant departments take preventive measures to reduce casualties and economic losses caused by landslides. Therefore, it is necessary to carry out LSE in the Yichang section of the Yangtze River Basin.

With the development of 3S technology, extensive data mining, and artificial intelligence, these technologies have been widely used in all walks of life [12–17]. LSE combined with data mining technology can significantly improve the efficiency of data acquisition, analysis, and processing and quickly and accurately establish a practical set of influencing factors to promote the efficient application of model methods [18,19]. Research on LSE is gradually divided into three categories: traditional regression analysis, machine learning and its integration method, and deep learning [20–22]. The traditional regression analysis methods mainly include frequency ratio, exponential entropy model, landslide density, logical regression, evidence weight, and Fisher discriminant analysis [23,24]. These methods generally use the landslide list as the prediction variable and establish a statistical regression model to predict the probability of landslide occurrence. However, there is a certain degree of subjectivity in factor selection and weight (or other parameters) allocation, so this method relies on expert experience to a certain extent [25]. Traditional machine learning methods and ensemble techniques mainly include Artificial Neural Networks, Random Forest (RF), Support Vector Machine, Classification, and Regression Tree Bag Algorithm [26–28]. These methods have essential similarities in selecting key influencing factors, reducing the influence of highly correlated factors on model generalization ability. In addition, these methods can support the comprehensive analysis of various influencing factors and better depict the nonlinear correlation between influencing factors and landslide susceptibility [29,30]. Therefore, they can achieve relatively high LSE accuracy. The most widely used deep learning methods include Convolutional Neural Networks (CNN) and Support Vector Machine, RF and Logistic Regression, CNN and Support Vector Machine, RF or Logistic Regression, Recurrent Neural Networks, CNN and Spatial Explicit Deep Learning Neural Networks [31–33]. Compared with traditional machine learning methods, deep learning has a more complex structure, which is more competitive in describing complex nonlinear problems [34]. In addition, due to solid learning strategies, deep learning can obtain better generalization ability than traditional machine learning.

However, in the LSE study, there is a problem that the accuracy of model training is susceptible to sample quality [35,36]. The samples include landslide samples with very high susceptibility or probability of 100% and non-landslide samples with shallow susceptibility or probability of 0 [37]. Landslides samples can be determined by field investigation and remote sensing interpretation of the landslide that has occurred, which is authentic. However, most non-landslide samples are randomly or subjectively selected, and the selected non-landslide grid cells cannot be well guaranteed to be "non-landslide" with high probability. Therefore, the correct selection of non-landslide samples is essential in ensuring sample quality and model accuracy [38]. Non-landslide samples are generally selected by manual screening and statistical methods, including random, buffer, slope, clustering. [39,40]. The random method is to generate non-landslide samples randomly outside the landslide range [41]; like the random method, the buffering method buffers the landslide and generates non-landslide samples in the buffer [42]; the slope method sets the slope threshold to generate non-landslide samples in areas less than the threshold [43]. However, the non-landslides selected by these methods are only subjective speculation or random selection of non-landslides, which cannot guarantee the low susceptibility of the selected non-landslides. Compared with the first three methods, the clustering method pre-classifies the study area according to certain rules and selects non-landslide samples in areas with extremely low susceptibility with high rationality [44]. The non-landslide samples selected by the clustering method are not very close to the landslide samples in space, which improves the quality of the non-landslide samples, avoids the over-fitting of the susceptibility model, and improves the evaluation accuracy.

To sum up, in order to overcome the lack of accuracy in the selection of non-landslide samples of landslides and to take into account a large number of influencing factors, comprehensive sources, and high dimensions, we have carried out three aspects of research on LSE: (1) Non-landslide samples selection based on DEC unsupervised learning; (2) Establishment of a landslide susceptibility evaluation model based on SE-CapNet deep learning technology; (3) Based on the analysis of the established LSE model and driving mechanism, mid- and long-term prediction of landslide susceptibility to rainfall under global change. Figure 1 shows the technical route of this paper.

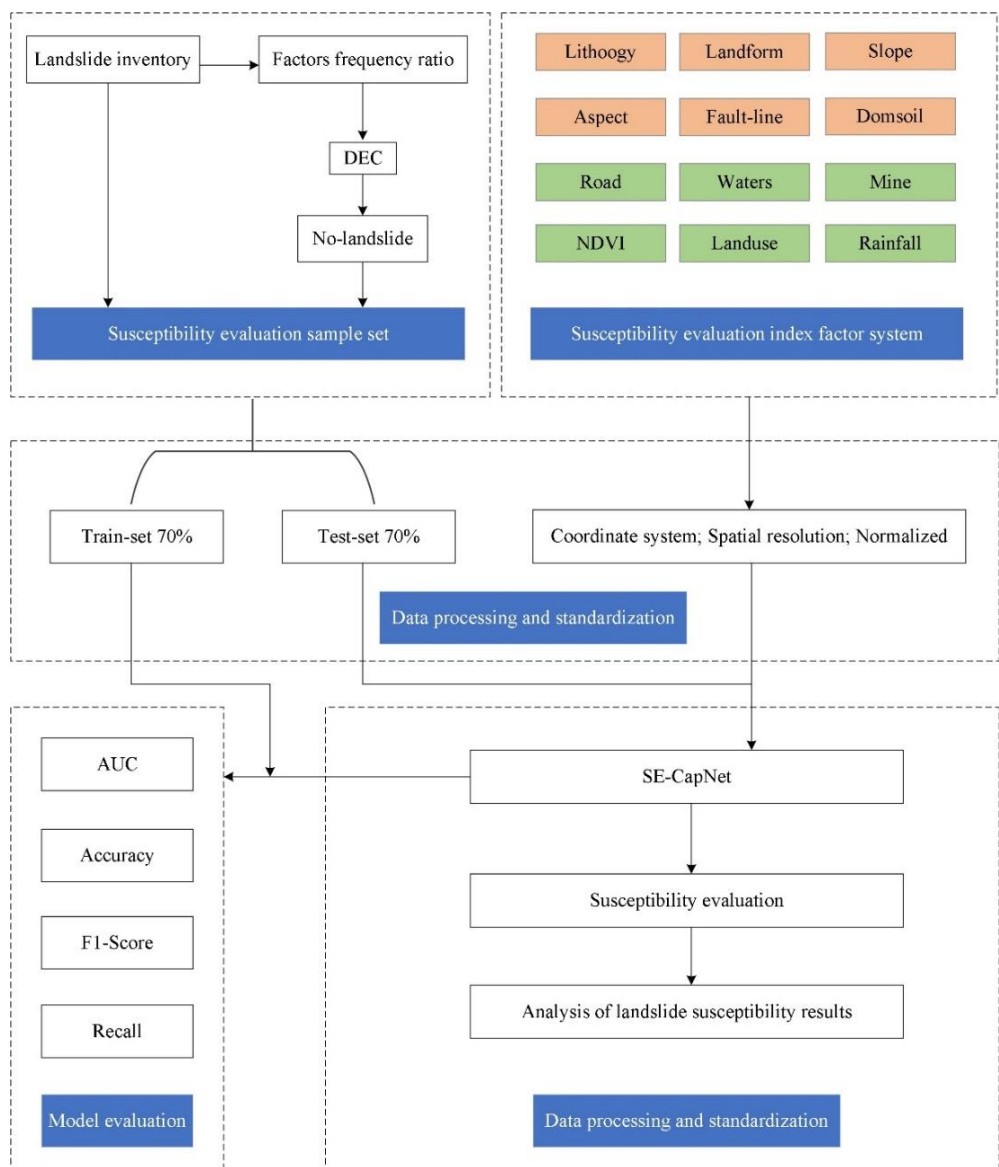

**Figure 1.** Technical Route.

## 2. Study Area and Data

### 2.1. Study Area

Mountainous and hilly areas in the Yangtze River Basin, interlaced plains and lakes, rich and varied topography, and abundant water resources are the basis for our survival and economic development [45]. Yichang is located in the junction of the middle and lower reaches of the Yangtze River at the lower end of the Three Gorges Reservoir area (Figure 2). The famous Three Gorges Dam is built in Yichang City [46,47]. Therefore, the research on resource protection and geological disaster prevention in the Yichang area of the Yangtze

River Basin has significant practical value. The high mountains, hills, and plains in the Yichang area of the Yangtze River Basin have complex geological conditions and frequent geological disasters. It is one of the areas with severe geological disasters such as landslides, debris flow, collapse, and ground collapse in Hubei Province [48,49]. According to statistics, about 599 geological disasters occur in the study area [50,51]. Frequent geological disasters seriously threaten people's lives and property safety and significantly affect social stability and local economic development [52]. Therefore, it is of great significance to strengthen the monitoring and early warning research of the geological disaster susceptibility evaluation in the Yichang section of the Yangtze River Basin if we are to protect people's life and material property from loss and the safety of the Three Gorges Reservoir area.

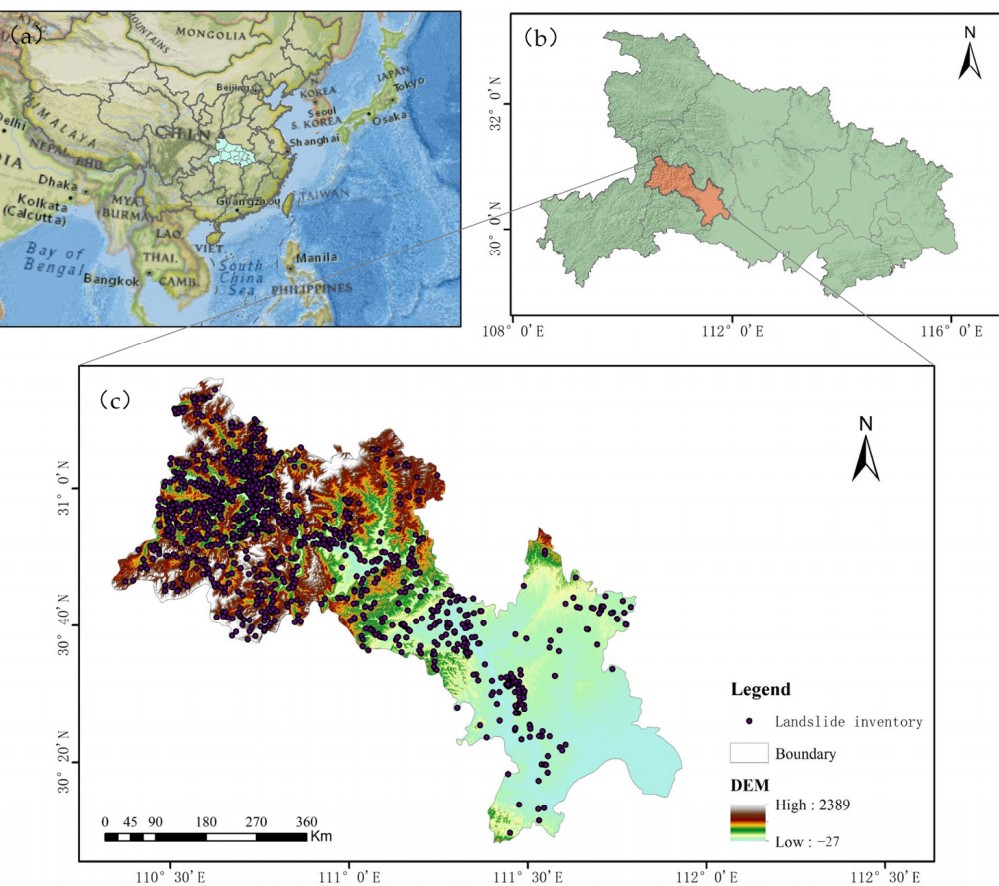

**Figure 2.** Geographical location of the study area. (**a**) Hubei Province in China; (**b**) Study area in the Hubei Province; (**c**) Yangtze River in the Yichang City.

*2.2. Data*

2.2.1. Landslide Inventory

As the first step of LSE, landslide investigation is necessary for modeling [53]. We used the global landslide catalog (GLC) opened by NASA as the data source, which can be downloaded and used free on Cooperative Open Online Landslide Repository (COOLR). The landslide catalog included 7556 records of training, testing, and prediction data. The data existed in the form of landslide points from 1996 to 2019. Among them, the landslide data in the Yichang section of the Yangtze River Basin contain 1270 records, and each record contains fields such as disaster type, county, township, village group, inducement, and area. Figure 2c shows the distribution of landslide data in the Yichang section of the Yangtze River Basin.

### 2.2.2. Data Sources

LSE predicts the probability of geological disasters in an area according to the historical disaster data given in the list. The data used usually include a variety of environmental factors and historical records of geological disasters [53,54]. Therefore, studying various influencing factors of landslide genesis is also necessary for LSM research. Based on previous LSM studies in the Yangtze River Basin, we use elevation data, primary geological and topographic data, and primary geographic data to produce LSE factors [55–57]. The ground resolution of the Landsat-8 remote sensing image is up to 15 m, and its wavelength range can be from 0.43 μm to 2.29 μm, including nine bands [58]. This paper mainly extracts soil vegetation index, land use, road, and water distance index factors based on Landsat-8 remote sensing image. Primary geological data from the National Geological Database is used to draw stratigraphic lithology and the geological structure fault index factor. Topographic data come from the free DEM products with a ground resolution of 12.5 m provided by Japan's Advanced Land Observing Satellite (ALOS), which extracts slope, aspect, and topographic factors. Table 1 shows the data in this paper.

**Table 1.** Details of the dataset used for LSE.

| Data Name | Data Source | Resolution | Purpose |
| --- | --- | --- | --- |
| GF-1 | Natural Resources Satellite Remote | 2 m | |
| GF-2 | Sensing Cloud Service Platform | 1 m | Landslide Inventory |
| Google Earth | Local Space Viewer | 2 m | |
| Landsat-8 | USGS | 30 m | Extraction of Vegetation Index, Road, Water System, Land Use. |
| Fundamental terrain data | NASA | 30 m | Extraction of topography, slope, and aspect. |
| Fundamental geological data | National Geological Archives of China | — | Draw stratigraphic lithology, geological disasters, and geological structure. |
| Fundamental geographic data | China Meteorological Data Network and Hubei Provincial Geological Survey | — | Precipitation and mine data sources. |
| Administrative division data | Global Administrative Division Database | — | Extraction of administrative boundaries. |

### 2.2.3. Indicator System

The geological conditions of the Yichang section in the Yangtze River Basin are complex, and the occurrence of landslides is affected by many factors. By analyzing the development law, temporal and spatial distribution characteristics of landslide disasters in the Yichang section of the Yangtze River Basin, it is known that the landslide disasters in the study area are mainly caused by the interaction of triggering factors such as topography and geomorphology conditions, vegetation coverage and human activities. According to previous research and survey results, 12 factors were selected from five categories: topography, introductory geology, hydrological and soil conditions, surface coating, and disaster-causing factors (Figure 3). Figure 4 shows a thematic map of the 12 factors in the study area.

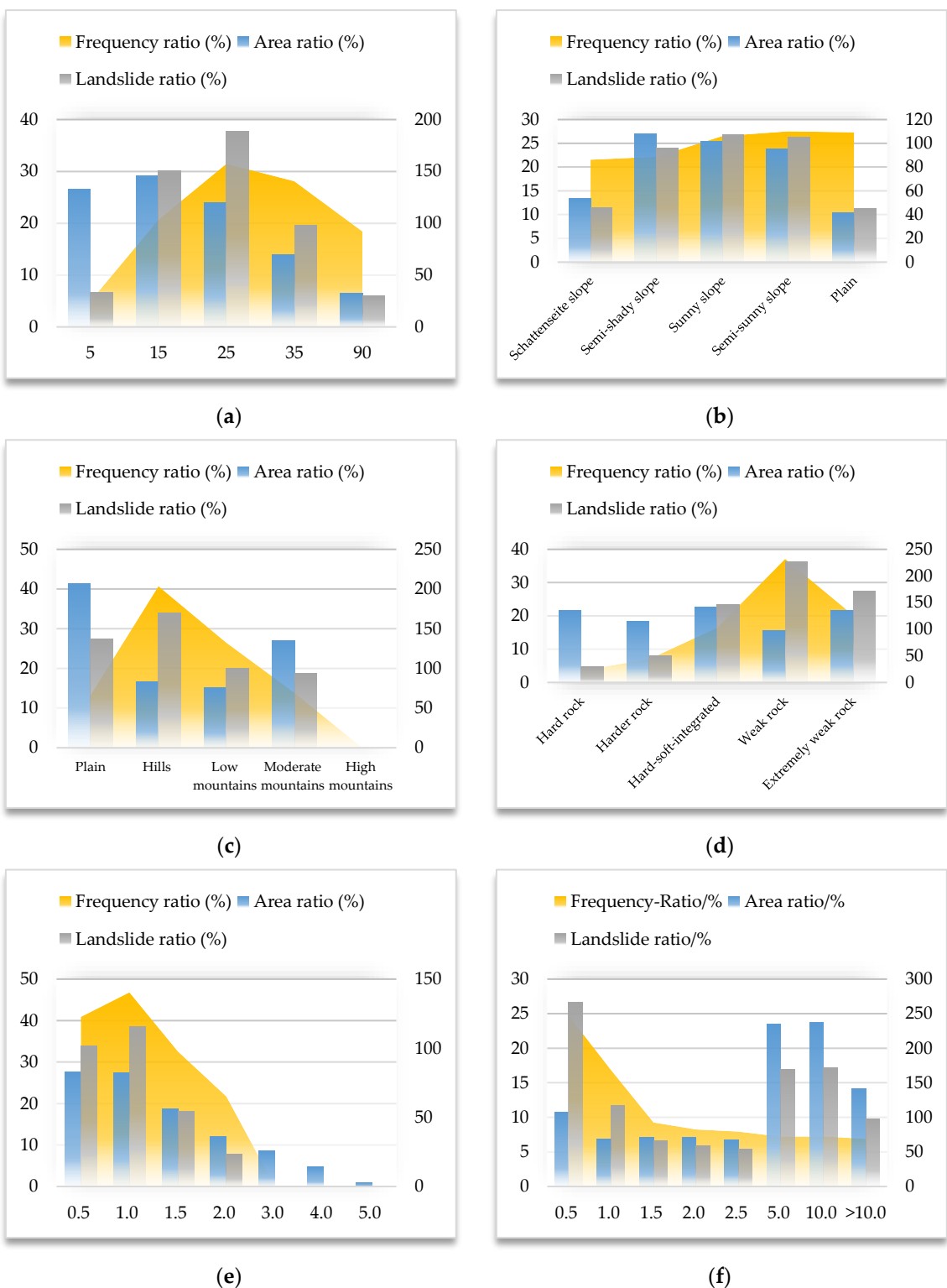

**Figure 3.** *Cont.*

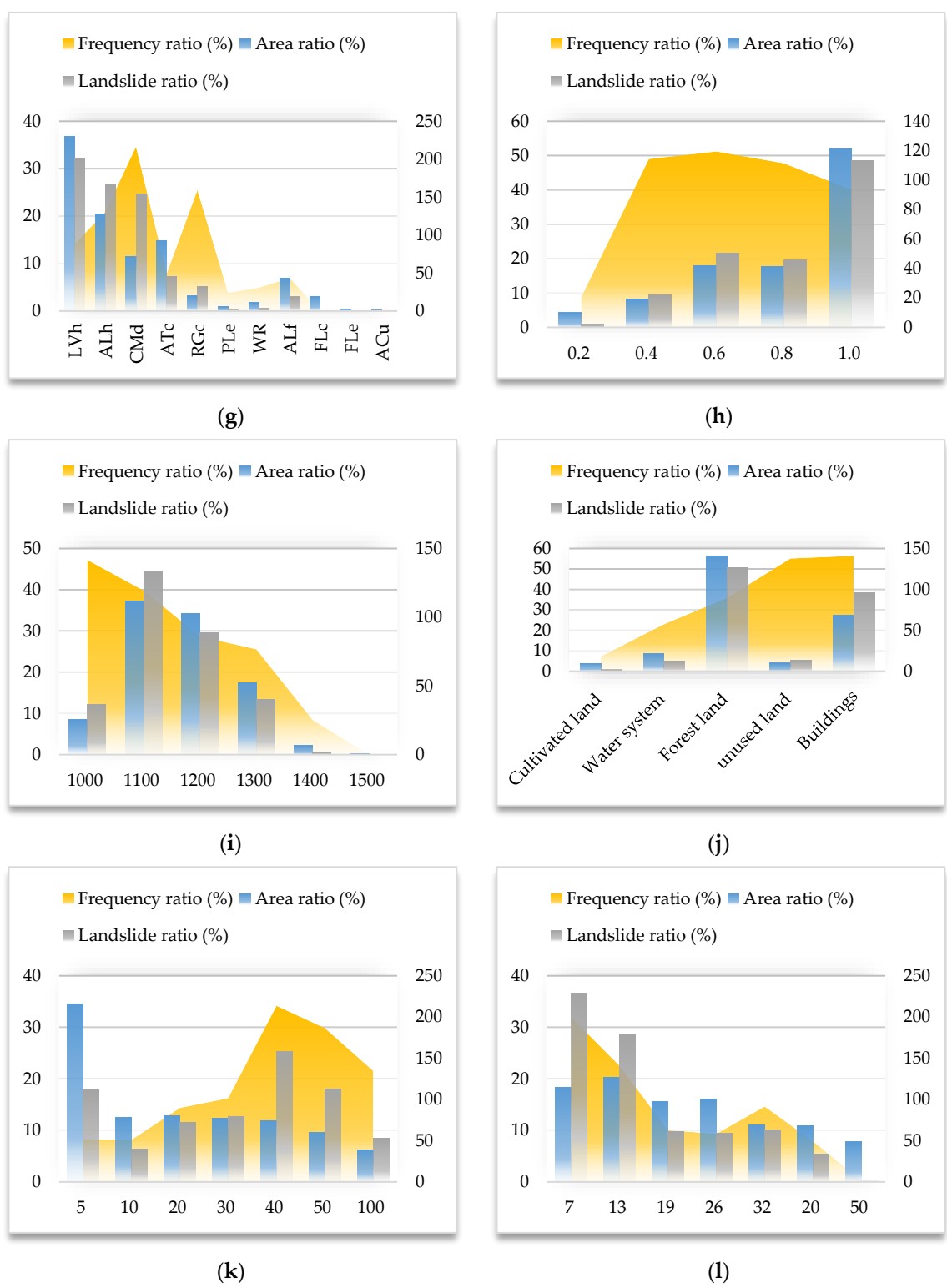

**Figure 3.** Analysis of influence factor and landslide frequency ratio. (**a**) Slope. (**b**) Aspect. (**c**) Landform. (**d**) Lithology. (**e**) Distance from fault. (**f**) Distance from water. (**g**) Domsoil. (**h**) NDVI. (**i**) Rainfall. (**j**) Landuse. (**k**) Distance from road. (**l**) Distance from mine. Where, the Landslides ratio is the gray histogram, indicating the proportion of landslides in each factor interval to all landslides; the Area ratio is the blue histogram, indicating the proportion of the area of each factor interval to the total area; the Frequency ratio is the yellow line graph, indicating the ratio of landslide proportion to area proportion in each factor interval.

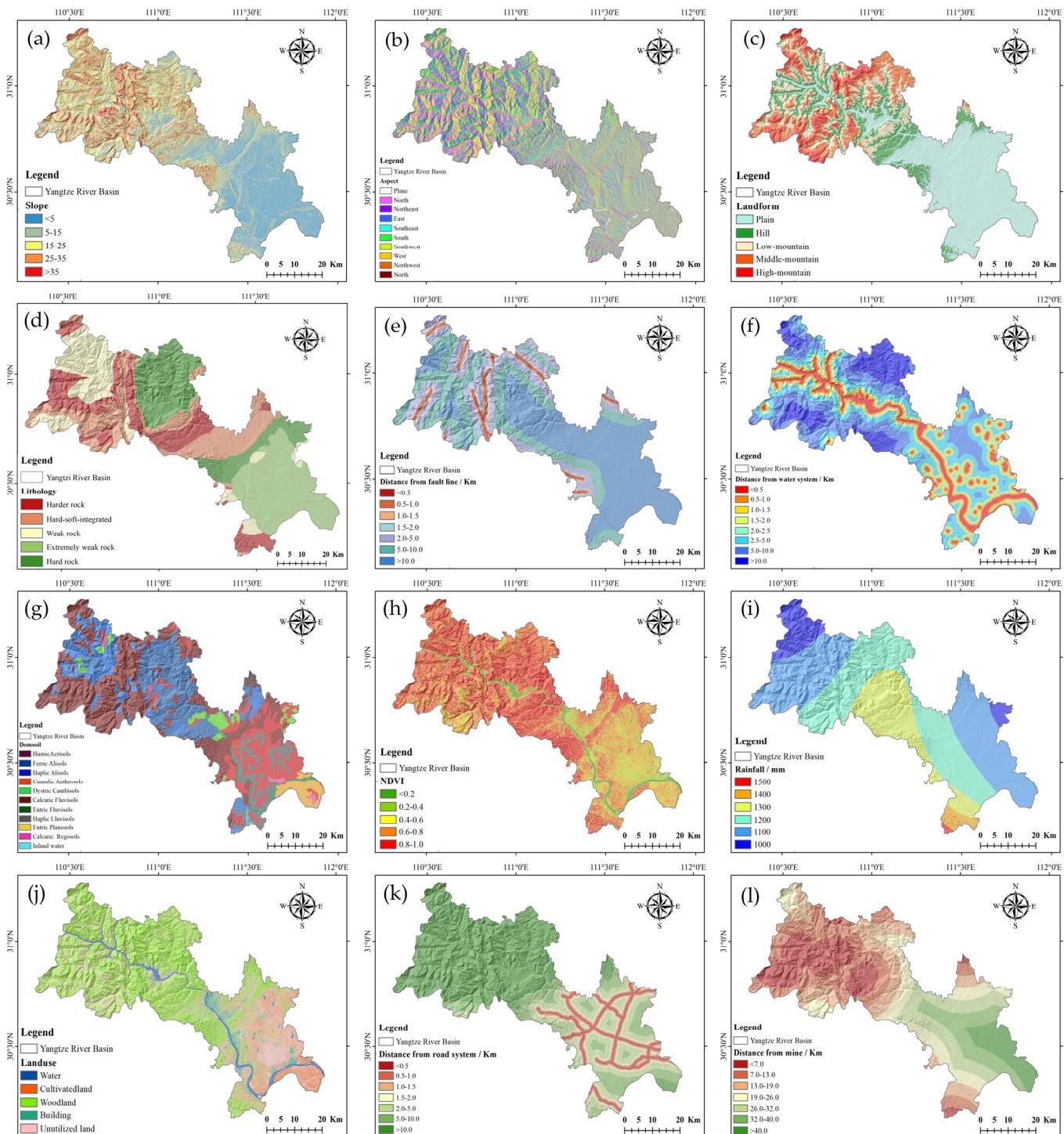

**Figure 4.** Thematic map of factors. (**a**) Slope; (**b**) Aspect; (**c**) Landform; (**d**) Lithology; (**e**) Distance from fault; (**f**) Distance from water; (**g**) Domsoil; (**h**) NDVI; (**i**) Rainfall; (**j**) Landuse; (**k**) Distance from road; (**l**) Distance from mine.

(1)    Topography

The slope is considered the critical topographic factor that directly affects the slope stability [59]. The slope will affect the seepage process and stress field distribution. The statistical relationship between slope and landslide in the study area is shown in Figure 3a. About 80% of landslides are distributed in the range of 15–35°, and the frequency ratio is

more significant than 100%, indicating that the landslide mainly occurs on the slope with medium slope.

Aspect is another crucial factor affecting the development of landslides, which will affect rainfall leakage and runoff and the absorption of solar radiation, thereby indirectly affecting the occurrence of landslides [60]. The slope direction of the study area is shown in Figure 3b. The frequency ratios of the sunny slope, semi-sunny slope, and flat slope are all greater than 1, indicating that the slope direction interval has a specific effect on landslide occurrence.

The landslide development is closely related to landform, and their relationship is shown in Figure 3c. The hilly area of the study area is the smallest, but the landslide disaster point is about 35%, and the frequency ratio is about 2. It shows that hills have the most significant impact on landslide development in the topography of the study area.

(2)    Geological factors

Formation lithology plays an essential role in developing landslides, which mainly affects the bedrock's physical and mechanical properties and is an essential internal factor for landslide disasters [61]. The strata in the study area are exposed from Paleozoic to Quaternary, and the lithology can be summarized as hard rock, harder rock, hard-soft-integrated, weak rock, and extremely weak rock (Figure 3d). Spatial statistical analysis of disaster points and lithological layers shows that landslides are developed in all strata, but weak rock, and extremely weak rock and other low-strength strata are the most developed. The development density of hard-soft-integrated is second, and the distribution of landslides in hard rock is tiny.

The distance from the fault is also a common geological factor for LSE. The fractured zone formed by the fault control has a soft surface or weak zone locally, and the rock and soil strength are low and easy to be weathered and denuded. In particular, there are cut slopes caused by artificial activities at the fractures, which are prone to landslide disasters under the action of rainfall, weathering, and erosion. Therefore, the area close to the fault fracture zone can easily become the most developed area of regional geological disasters and hidden dangers within a specific range. Figure 3e shows that the area proportion and landslide proportion of the study area from the fault distance between 0.5–1.0 km are the highest, and the frequency ratio is greater than 1. We see that the closer the distance from the fault, the higher the probability of landslide occurrence.

(3)    Hydrological and soil conditions

The Yangtze River Basin is rich in water systems, and the river is densely distributed. When the river erodes at the bottom of the slope, the bottom of the slope will be filled with pore water, resulting in a decrease in slope stability. Therefore, the distance from the water system is also an essential factor for LSE in this study area. The results in Figure 3f further show that when the distance to the water system is less than 0.5 km, the maximum frequency ratio is 2.47, followed by 0.5–1.0 km, and the frequency ratio continues to decrease with the increase of the distance to the water body.

Due to the different domsoil, the characteristics of soil water content and viscosity are also different, and so the friction degree between the affected body and the surface of different soil types is significantly different. Figure 3g shows the soils in the study area, which are mainly summarized into eleven categories, including Haplic Luvisols (LVh), Haplic Alisols (ALh), Dystric Cambisols (CMd), Cumulic Anthrosols (ATc), Calcaric Regosols (RGc), Eutric Planosols (Ple), Inland Water (WR), Ferric Alisols (Alf), Calcaric Fluvisols (FLc), Eutric Fluvisols (Fle), and Humic Acrisols (Acu). Figure 3g shows that the area ratio and landslide ratio of the LVh are the largest, which were 36.7% and 32.3%, respectively, but the frequency ratio is only 0.8. The highest frequency ratio of RGc is 1.5, and the next highest frequency ratio of ALh is 1.3. Although the area of the two soil types and the proportion of landslides are not high, the occurrence of landslides is more frequent, indicating that it has an important impact on the development of landslides.

(4)　Surface coverage

The vegetation index has a specific influence on slope stability, and its roots can improve the shear strength of the soil. Leaf transpiration can promote groundwater discharge and soil slope protection. Therefore, generally, the denser the vegetation, the better the slope stability. It can be seen from Figure 3h that the area of NDVI in the study area was the highest between 0.8 and 1, accounting for about 50%. The landslide frequency ratio of NDVI < 0.6 is the highest, showing that landslide probability is higher in the low vegetation area.

(5)　Disaster-causing factors

➢　　Rainfall

Rainfall is one of the most critical factors causing slope landslides. Since the study area is in the subtropical climate zone with a mild and humid climate, abundant rainfall, continuous rainfall, and other related functions are the main causing factors for landslide development in this area. Based on the statistical analysis in Figure 3i, the average annual rainfall is the highest in the range of 1000–1300 mm, and the accumulation of rainfall will aggravate the occurrence of landslides.

➢　　Human engineering activities

With the development of the social economy, the scale and intensity of human activities have become larger and larger, and their speed has exceeded the development of natural geology, becoming a vast force affecting the development of landslides. Human activities such as urban construction, highway reconstruction, and mineral exploitation in the Yichang section of the Yangtze River Basin are directly or indirectly related to landslide development. Therefore, we used land use, distance from roads and distance from mines in LSE.

Land use types in the Yichang section of the Yangtze River Basin can be divided into five categories: cultivated land, water system, forest land, unused land, and buildings (Figure 3j). The statistical results show that the number of cultivated land landslides is the least, and the frequency ratio is only 0.18. The highest proportion of landslides in unused land is about 50%, and the frequency ratios to buildings are 1.37 and 1.41, respectively. The frequency of landslides in a specific area is the highest, indicating that landslides in woodland and buildings are the most developed.

The road is a kind of widely distributed artificial building. Road construction will interfere with earthwork to a certain extent and indirectly lead to slope instability. The frequency of landslides between different distances from the road in the study area is shown in Figure 3k. According to the statistical results, the landslide accounts for a relatively high proportion at 5 km and 30–50 km from the road. However, when it is farther away from the road, the frequency ratio does not appear to be significantly reduced, indicating that the road distance in the study area is not apparent for landslide development, which can be used as an indirect adjustment factor.

The Yichang section of the Yangtze River Basin is rich in mineral resources generated by intense geological activities in the geological history period, accounting for 62% and 51.2% of the discovered mineral types in the whole province and the whole country, respectively. The risk of geological disasters caused by mining is severe. The landslide frequency of the distance from the mine in the study area is shown in Figure 3l. We see that the landslide occurs most frequently in the study area with a distance less than 7 km from the mine, and the frequency ratio is as high as about 2.0, followed by the relatively developed within the range of 7–13 km. The smaller the distance from the mine, the lower the landslide frequency is, indicating that mining has played a crucial role in landslide development.

## 3. Methodology

### 3.1. Non-Landslide Samples Selection Network Based on DEC

To overcome the limitations of large randomness and strong human dependence in the selection of non-landslide samples in traditional methods, we use a typical unsupervised DEC to select non-landslide samples. DEC is composed of stacked autoencoders and soft allocation models [62,63]. DEC can learn latent feature representation and clustering allocation as a joint model of dimensionality reduction and clustering. Compared with K-means, Space Optimal Aggregation Model (SOAM), Density-Based Spatial Clustering of Applications with Noise (DBSACAN), and other clustering methods, the stacked automatic encoder architecture helps to reduce redundant information and high-dimensional noise, which is convenient to solve the high-dimensional problem of multi-source heterogeneous data [64]. As shown in Figure 5, the unsupervised DEC is composed of a fully connected stacked automatic encoder (SAE) and soft allocation model of T distribution measurement, which is trained by matching soft allocation with target distribution.

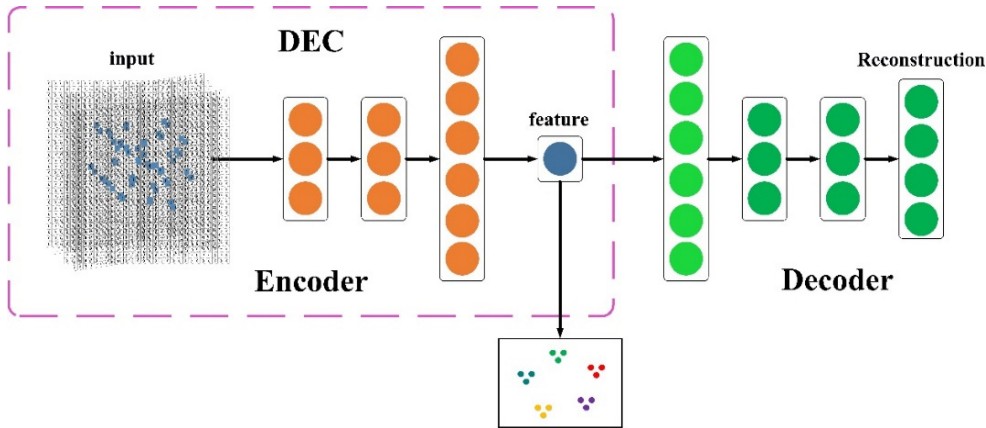

**Figure 5.** The framework of DEC.

The specific process of non-landslide sample selection is as follows. First, the DEC neural network is used to cluster the landslide susceptibility in the study area systematically. Then the non-landslides are selected from the extremely low prone areas of the preliminary clustering results to ensure that the selected grid units have a very low probability of landslide occurrence. Finally, LSE model based on SE-CapNet network is constructed based on the sample data set composed of non-landslides classified and selected.

### 3.2. Capsule Neural Network Based on SENet

CNN can maintain the spatial invariance of input data or tolerate small spatial changes through operations in the convolution and pooling layers [65]. Nevertheless, it loses detailed information to a certain extent, especially for remote sensing images with rich detail texture information. Aiming at the problem of feature information loss and poor generalization ability caused by CNN classification of remote sensing images, we use an improved Capsule Neural Network model (CapNet) based on Squeeze-and-Excitation Model (SENet) model. CapNet used capsule to replace neurons in CNN, so that the network can retain detailed attitude information and hierarchical spatial relationship between objects and make up for the defects of CNN [66]. At the same time, the SENet model can obtain the importance of each feature channel through learning in the training process of CapNet, and it will automatically improve the valuable features according to the importance and suppress the features that are not very useful for the current task, to obtain more accurate susceptibility results [67]. As shown in Figure 6, the SE-CapNet network structure consists of the SENet feature extraction part, the Capsule neuron part, the dynamic routing part, and the classification capsule network part.

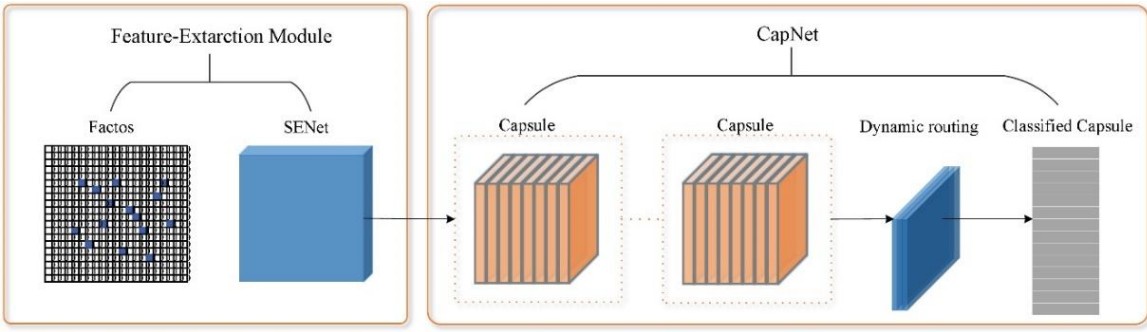

**Figure 6.** The framework of SE-CapNet.

(1) SENet Feature Extraction Network

SENet won the first place in the ImageNet2017 classification task. It adds processing between adjacent two layers, making the information interaction between channels possible and further improving the accuracy of the network [68]. As shown in Figure 7, SENet mainly consists of two parts. In the Squeeze part, the global vision is obtained by increasing the sensory area while reducing the dimensionality of the image data. The Excitation section adds a fully connected layer to predict the importance of each channel, get the importance of different channels, then act on the compressed feature map, and finally input the feature map into the CapNet network.

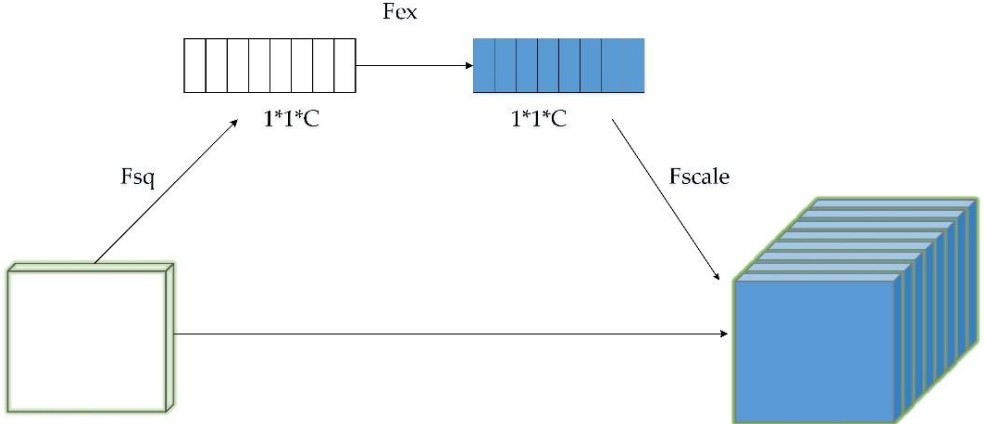

**Figure 7.** The framework of SENet.

(2) Capsule neurons

The purpose is to fuse the features extracted from the previous convolution layer and input them into the dynamic routing layer. The essence of the capsule neuron is similar to the convolution layer, and each capsule layer corresponds to different input layers. In the inner layer, the m-dimensional space of the input is mapped to the n-dimensional space of the output by weight matrix processing.

(3) Dynamic routing

The dynamic routing algorithm mechanism measures the similarity between input and output by calculating the dot product of the input and output of the capsule and then updates the routing coefficient $b_{ij}$ of the neural network according to the dot product value [69]. Firstly, all $b_{ij}$ is initialized to 0, and iterative calculation is started. Each iteration first calculates the $c_{ij}$ value by softmax function and then combines $U_{ij}$, $w_{ij}$, and $c_{ij}$ to do linear summation to obtain $S_j$. Then $S_j$ is input into the activation function Squash to obtain $V_j$. Finally, $\hat{U}_{j|i}$ and $V_j$ is used to update the $b_{ij}$ value. After all, calculations start the next iteration and use three iterations for best practices in practice. The update expression for $b_{ij}$ is shown in Equation (1):

$$b_{ij} \leftarrow b_{ij} + \hat{U}_{j|i} V_j \tag{1}$$

The coupling coefficient $c_{ij}$ is updated through the dynamic routing algorithm, but the other convolution parameters of the entire network and $w_{ij}$ in the capsule need to be updated according to the loss function.

(4) Classification capsule

The multidimensional vector output by the classification capsule can be reduced in the fully connectional layer and converted from vector to scalar. In the capsule network, the fully connectional layer will integrate all the features obtained previously and enhance the robustness of the network. Multiple fully connectional layers can also increase the nonlinear expression ability of the network, which is more conducive to network learning. However, the number of fully connectional layers and the number of neurons in each layer will increase the number of parameters in the network and lead to over-fitting. Therefore, we set the fully connectional layer to three layers, and the number of neurons in the last layer should be consistent with the number of different categories in the classification results of the selected dataset. The vector value from the classification capsule layer is converted into scalar data in the fully connectional layer after the operation. After integration, it is mapped to n classification nodes to realize the spatial transformation of features and output the classification results.

### 3.3. Precision Evaluation Indicators

Usually, the magnitude of landslide and non-landslide samples is not the same. For such unbalanced learning problems, it is challenging to obtain ideal results by using classification accuracy alone to evaluate the model's performance, leading to high accuracy and low recall rate [70,71]. Therefore, in this study, we used the four statistical indicators of accuracy, precision, susceptibility, and specificity, as well as the Receiver Operating Characteristic (ROC) curve and the curve below, and Area Under Curve (AUC) based on susceptibility and specificity to evaluate the performance of LSE model [72–74]. Among them, sensitivity is the proportion of landslide samples in the correct classification to all landslide samples; specificity is the proportion of non-landslide samples in the correct classification to all non-landslide samples. Equations (2)–(5) shows how the indicator is calculated.

$$\text{Accurary} = \frac{\text{TP} + \text{TN}}{\text{TP} + \text{FP} + \text{TN} + \text{FN}} \tag{2}$$

$$\text{Precision} = \frac{\text{TP}}{\text{TP} + \text{FP}} \tag{3}$$

$$\text{Sensitivity} = \frac{\text{TP}}{\text{TP} + \text{FN}} \tag{4}$$

$$\text{Specificity} = \frac{\text{TN}}{\text{FP} + \text{TN}} \tag{5}$$

TP (True Positive) and TN (True Negative) are the numbers of grids correctly classified, while FP (False Positive) and FN (False Negative) are the numbers of grids wrongly classified [75]. ROC curve is often used to evaluate the performance of diagnostic signals and prediction models, with sensitivity as the *Y*-axis and Specificity as the *X*-axis. AUC represents the ability of the model to predict landslide and non-landslide grids. When its value is 1, it represents the perfect model, while 0 represents the invalid model.

### 4. Results

### 4.1. Training Based on Integrated Deep Learning Algorithm

4.1.1. Non-Landslide Samples Set Selection

In this paper, the frequency ratios of 12 environmental factors in the index system are standardized as input variables of DEC, and the preliminary classification results are shown in Figure 8. The natural breakpoint method was used to classify the susceptibility results of

DEC landslide, and five categories were obtained: extremely high, high, medium, low, and extremely low. The frequency ratio statistical results are shown in Table 2. It can be seen from Table 2 that the proportions of each grade of LSE are extremely low, low, medium, high and extremely high respectively. The extremely low susceptibility area accounts for 18.07% of the total area of the study area but only 8.46% of the total number of landslide grid units. At the same time, the extremely high and high landslide susceptibility areas in the study area contain about 62.11% of the landslide grid units, and the frequency ratio of the extremely high and high landslide susceptibility areas accounts for 59.52% of the total frequency ratio. The validity of the DEC-based LSE results has been shown.

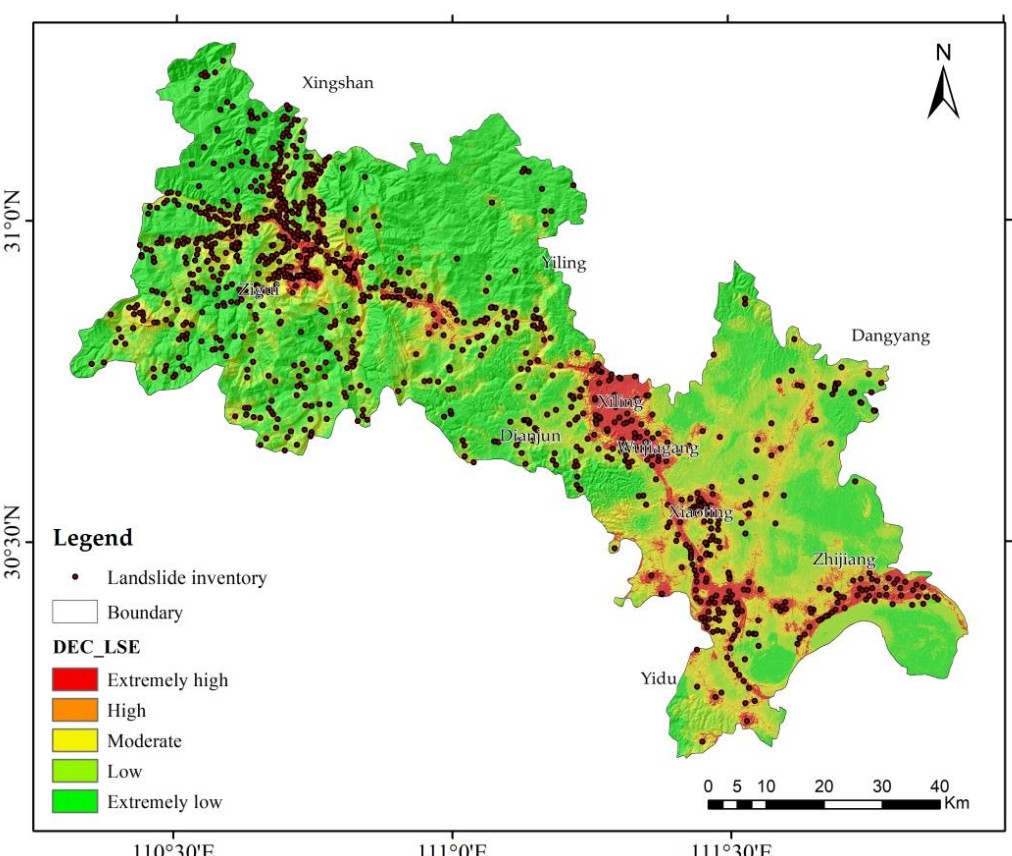

**Figure 8.** LSE result based on DEC.

**Table 2.** DEC-Result.

| LSE | Area Ratio/% | Landslide Ratio/% | Frequency Ratio/% |
| --- | --- | --- | --- |
| Extremely high | 29.24 | 32.35 | 110.64 |
| High | 15.39 | 29.76 | 193.37 |
| Moderate | 25.11 | 19.41 | 77.30 |
| Low | 12.40 | 10.25 | 82.66 |
| Extremely low | 18.07 | 8.46 | 46.82 |

Then the non-landslide grid cells were selected from the extremely low area of DEC preliminary LSE results. To verify the rationality of the non-landslide sample selection, some sample images were randomly selected and projected onto Google Earth (Figure 9). Figure 9($a_1$) and Figure 9($a_2$) show that landslide samples are generally close to roads and water systems, while Figure 9($a_3$) shows that landslides develop more widely in areas with frequent human activities. It can be seen from the image of the example in Figure 9b that the non-landslide samples are evenly distributed in urban areas, mountainous areas, water systems, and so on. With gentle terrain, dense vegetation, and less human activities.

Combined with prior knowledge, topographic data and hydrological data, it can be seen that the locations of non-landslide samples and landslide samples are quite different, indicating that the selection method of non-landslide samples based on DEC clustering is reasonable.

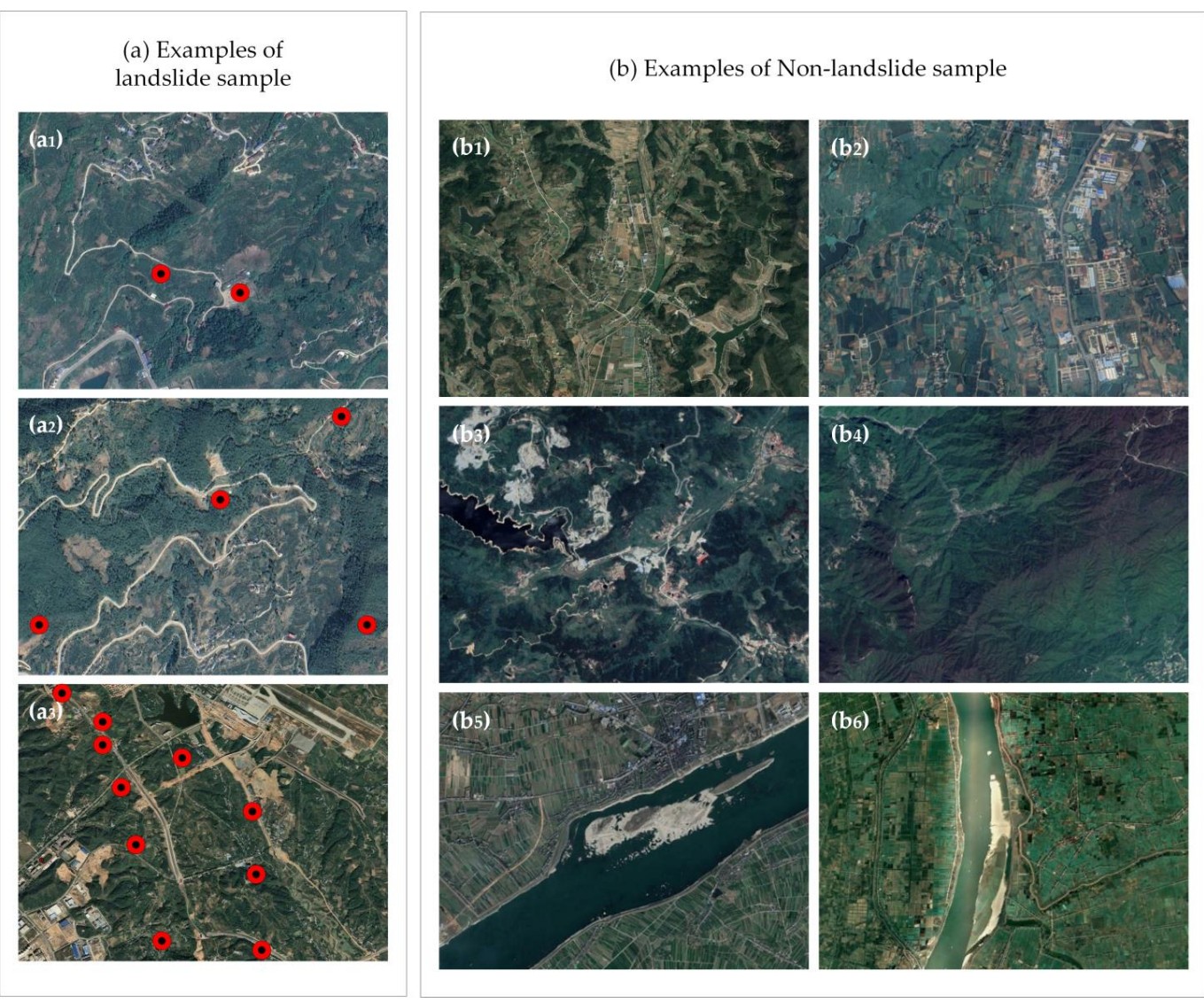

**Figure 9.** Examples of non-landslide sample. (**a**) Examples of landslide sample; (**b**) Examples of Non-landslide sample; (**a₁**–**a₃**) Landslide samples from different regions; (**b₁**,**b₂**) Landslide samples from different towns; (**b₃**,**b₄**) Landslide samples from different mountains; (**b₅**,**b₆**) Landslide samples from different mountains.

4.1.2. Environment and Training Parameters

In this study, 5621 landslide points in the whole province were randomly selected as positive samples, and the training samples of SE-CapNet model were constructed by using the selected 10,242 non-landslide points. At the same time, the study area was selected as the verification sample of the model.

The experiment is carried out in a 16 GB Windows 64bit operating system. The intel CORE i59th Gen CPU is configured, and the GeForce GTX 1650TI card is mounted. We select TensorFlow as a learning framework, mainly relying on libraries such as TensorFlow, Keras, OpenCV, and PIL. The specific experimental environment configuration is shown in Table 3.

**Table 3.** Experimental environment configuration.

| Hardware Device | CPU: Intel CORE i5 9th Gen<br>GPU: NVIDIA GeForce GTX 1650TI |
|---|---|
| System platform | Windows10 64-bit |
| Development environment | Python 3.6.5, TensorFlow-GPU 1.9.0, Keras 2.1.6 |
| Compile environment | Anaconda3, Jupyter |

*4.2. LSE Results from Integrated Deep Learning Algorithm*

4.2.1. Accuracy Assessment and Algorithm Comparison

To verify the effectiveness of the proposed method, experiments were carried out on the constructed landslide data set in RF, CNN, CapNet, and the SE-CapNet deep learning algorithm integrated with this paper. Table 4 lists the precision index values used to assist in evaluating the model's performance. From the table, it can be seen that in terms of sensitivity, the integrated algorithm in this paper obtains the maximum value (95.12%), followed by the convolution neural network CapNet algorithm (91.37%), CNN (89.02%), and RF (82.60%), which shows that the proportion of positive samples of landslide can be detected by this method is high. In terms of specificity, RF achieved the value (89.27%), followed by CNN (91.42%), CapNet (94.05%), and SE-CapNet (96.83%). In terms of accuracy and precision, the integrated model in this paper still achieves the maximum value: 96.06% and 96.82%. The proposed algorithm is superior to all models based on several important metrics, except that RF exceeds the specific value. It indicates that the method in this paper can meet the LSE.

**Table 4.** Comparison of precision index results.

| Methods | Accuracy (%) | Precision (%) | Sensitive (%) | Specificity (%) |
|---|---|---|---|---|
| SE-CapNet | 96.06 | 96.82 | 95.12 | 96.83 |
| CapNet | 93.30 | 94.29 | 91.37 | 94.05 |
| CNN | 91.57 | 92.36 | 89.02 | 91.42 |
| RF | 87.23 | 88.41 | 82.60 | 89.27 |

The ROC curve analysis of the model is shown in Figure 10. We use AUC value as the primary criterion for evaluating various models. The comparison of AUC values showed that all application models performed well in LSE (accuracy > 0.8). Compared with the other three methods, the AUC value in this paper is the highest, reaching 0.973, indicating that the method in this paper has the best performance in LSE. For CapNet and CNN, their performance is as good as ever, 0.965 and 0.931, respectively. RF also showed good generalization ability, and its AUC value was as high as 0.897. The method has a strong classification ability, but it is not stable because the variables are randomly selected and reclassified before each model training. The comparison results of the above indicators show that the deep learning integration scheme makes the model produce a diversified learning process, effectively reduces the generalization error caused by various preferences, and improves the prediction ability of the vulnerability evaluation model.

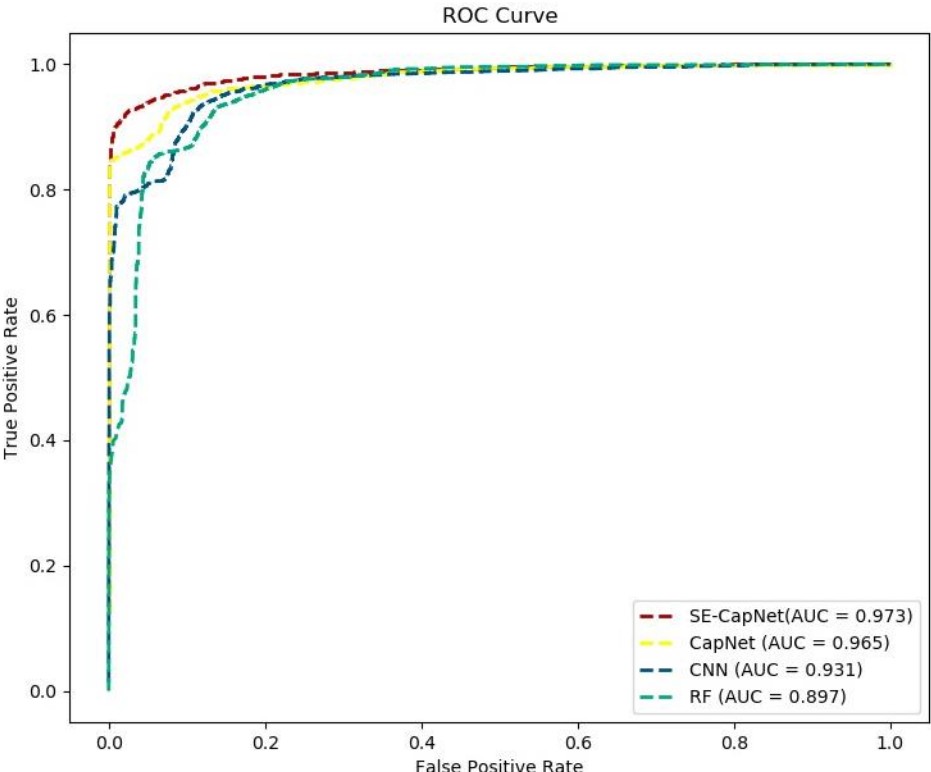

**Figure 10.** ROC Curve.

4.2.2. Verification and Algorithm Comparison

According to the trained model, the data of the whole study area is used as the model's input, and the final susceptibility results obtained by the integrated algorithm are shown in Figure 11d. Each grid point in the study area has the corresponding probability value of landslide susceptibility. The probability of landslide susceptibility is divided into five categories by using the natural breakpoint method: extremely low, low, medium, high, and extremely high. According to the evaluation result map, the area with high landslide susceptibility is consistent with the existing landslide area, mainly distributed in the Yangtze River Mainstream Region, and the landslide susceptibility on both sides of the watershed is generally high. The susceptibility evaluation results of RF, CNN, and CapNet are shown in Figure 11a–c. The graph shows that the regions with high susceptibility to the three methods are consistent with the distribution of known landslide points, which is consistent with the results of this method. However, the regions with high susceptibility in the final results of the above three methods are smaller than those of the artificial neural network.

At the same time, to further compare the performance of the four methods, the historical landslide disaster points in the study area are used to verify the results. The statistical table is shown in Figure 12. It can be seen from the table that 1270 landslide points in the known landslide points are located in the areas above the medium susceptibility predicted by the model, including 412 high susceptibility areas and 512 extremely high susceptibility areas, accounting for 73.05% of the total number of known landslides, and the susceptibility evaluation effect is good. Compared with this method, there are 471 landslide points in the high-prone areas of RF, accounting for 37.07% of the total landslides. There are 751 landslide points in the high-prone areas of CNN, accounting for 59.12% of the total number of landslides. There are 855 landslide points in CapNet, accounting for 67.35% of the total number of landslides. The above four methods have certain applicability in LSE, among which the method in this paper has the highest accuracy, indicating that the LSE model based on integrated deep learning has the best effect. This is because after the integration of SENet network, this method will obtain the importance of each feature

channel through learning in the training process, and automatically improve the valuable features according to the importance while suppressing the features that are not very useful for the current task. As a result, the extremely high-prone area is large. At the same time, the DEC is used to make the recognition rate of unbalanced landslide samples higher, so the final result accuracy is higher than other algorithms.

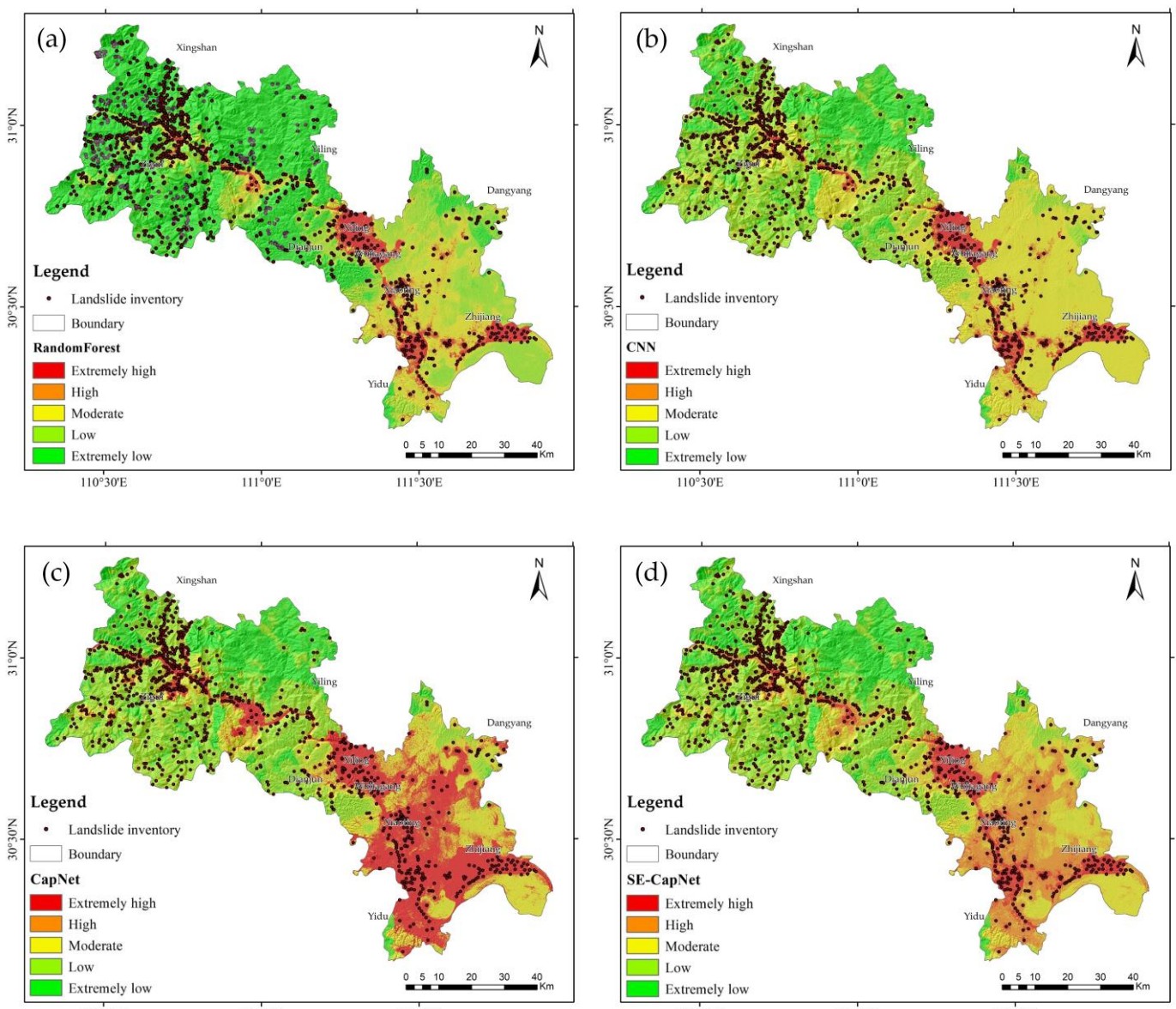

**Figure 11.** LSE Based on various methods. (**a**) RF based LSE result; (**b**) CNN based LSE result; (**c**) CapNet based LSE result; (**d**) SE-CapNet based LSE result.

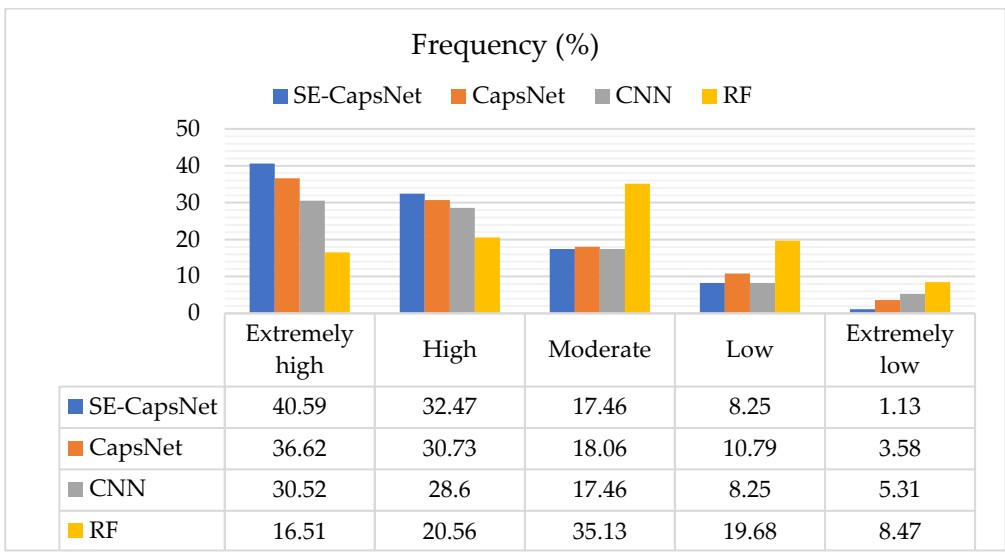

**Figure 12.** Frequency Based on Various Methods.

## 5. Discussions

### 5.1. LSE Results and Influencing Factors

5.1.1. Topography

(1) Slope: In the extremely low, low, medium, high, and extremely high-prone areas in the study area, the slope of different regions is affected by the slope of different regions, and the slope of different intervals is distributed in each grade. In this paper, the most significant proportion of slope intervals in each grade is taken as the main influencing factor (other factors are used in the same statistical method). The statistical analysis results are shown in Table 5. It can be seen from the table that the slope ranges of the high and extremely high-prone areas in this study area are 15–25° and 25–35°, and the slope ranges of the low and extremely low prone areas are >35° and <5°, which is consistent with the statistical characteristics that the landslide mainly occurs on the slope with the medium slope in the evaluation index.

**Table 5.** LSE results and influencing factors.

| LSE | Extremely High | High | Moderate | Low | Extremely Low |
|---|---|---|---|---|---|
| Area ratio | 14.67% | 29.47% | 30.24% | 17.29% | 8.33% |
| Slope | 15–25° | 25–35° | 5–15° | >35° | <5° |
| Aspect | West, Southwest, North | Northwest, Northeast | Southeast, East | Plane | South |
| Landform | Hill | Low mountains | Moderate mountains | Plane | High mountains |
| Lithology | Hard-soft-integrated | Weak rock | Extremely weak rock | Harder rock | Hard rock |
| Fault line | 0.5–1.0 km | <0.5 km | 1.0–1.5 km | 1.5–2.0 km | >5.0 km |
| Distance from water | <0.5 km | 0.5–1.5 km | 1.5–2.5 km | 2.5–10.0 km | >10.0 km |
| Domsoil | CMd, RGc | Alh, LVh | Atc, Alf, WR | Ple | FLc, Fle, Acu |
| NDVI | 0.4–0.6 | 0.2–0.4 | 0.6–0.8 | 0.8–1.0 | 0–0.2 |
| Rainfall | 1200–1400 mm | 1100–1200 mm | 1000–1100 mm | <1000 mm | 1400–1500 mm |
| Landuse | Building | Unutilized land | Woodland | Water | Cultivated land |
| Distance from road | 2.0–5.0 km | 5.0–10.0 km | 0.5–2.0 km | <0.5 km | >10.0 km |
| Distance from mine | <7.0 km | 7–13, 26–32 km | 13–26 km | 32–40 km | >40 km |

(2) Slope aspect: Slope aspect factors mainly control the degree of rock weathering, which is one of the parameters that cause the development characteristics of geomorphological differences. According to the statistical results, the areas with high and extremely high susceptibility are mainly affected by the West, Southwest, North, Northwest, and Northeast, indicating that the slope direction interval has a specific effect on landslide occurrence.

(3) Geomorphology: The study area in this paper generally forms three basic geomorphologic types, namely, mountains, hills, and plains. The mountains and hills are mainly distributed in the northwest of the study area, and the central and southeast of the study area are mainly plains. The development of landslides is closely related to topography. The hilly landform area in the study area is the smallest, but the proportion is the highest in the highly prone area to landslides, followed by low and moderate mountains.

### 5.1.2. Geological Factors

(1) Engineering rock group: The stratigraphic lithology in the geological environment is divided into five types according to the lithologic combination: hard-soft-integrated, weak rock, extremely weak rock, harder rock, hard rock. The extremely weak rock group in the study area is mainly distributed in the southeast, mainly composed of Quaternary clay and sandy clay, and the hard rock group composed of volcanic rocks and metamorphism is mainly distributed in the central and northern parts of the study area. After statistical analysis, the high-prone areas and high-prone engineering rock groups in the study area are mainly hard-soft-integrated and Weak rock, which are distributed in relatively weak or weak rock groups such as Quaternary clay, Cretaceous Glutenite and Devonian shale and limestone. In low and very low susceptibility areas, engineering rock groups are mainly harder rock and hard rock, and hard or hard engineering rock groups such as volcanic rock, metamorphic rock, dolomite, or limestone. The engineering rock group plays an essential role in developing landslides, mainly by affecting the physical and mechanical properties of bedrock and the accumulation body.

(2) Faults: The study area is located in the composite part of the southern section of the third uplift belt of the first-order structure of the Neocathaysian system and the Huaiyangshan type structure system in geological structure, and the fault structure is mainly developed in the northern and northwestern parts of the study area. The closer the distance from the fault, the larger the statistical area of the extremely high and high-prone areas is, indicating that the closer the fault, the more unstable is the rock, and the looser the soil, the higher is the possibility of a landslide.

### 5.1.3. Hydrology and Rainfall

(1) Hydrology: The Yangtze River is the main river in Yichang City. The river network is dense, and the water is abundant. The distance from the water body can be used to express the information about river development and river basin erosion, which is an essential factor in regional ecological stability. The extremely high and high-prone areas in this study area showed a high correlation with the distance from the river, and the area less than 0.5 km was the most prone, followed by the 0.5–1.5 interval. The greater the distance from the river, the lower the susceptibility.

(2) Rainfall: The rainfall index factor in the ecological environment is the crucial factor affecting the shear strength of engineering slope in evaluating geological environment carrying capacity. The rainfall in the study area varies from 1000 mm to 1500 mm, and the rainfall in the Northwest and Northeast is less than 1000 mm. The middle and south area is more prominent, about 1300 mm, and shows a trend of more in the middle and less on both sides. After statistical analysis, the areas with extremely high and high susceptibility in this study area are mainly distributed in areas more fabulous than 1200 mm.

### 5.1.4. Landcover

(1) NDVI: The results of LSE in the study area are approximately positively correlated with the distribution of NDVI. The NDVI index of high and high-prone areas is generally 0.2–0.6, and the NDVI index of low and very low prone areas is generally between B0.8–1.0 and 0–0.2.

(2) Soil types: The soil types in the study area are complex. In general, Dystric Cambisols (CMd) and Calcaric Regosols (RGc) are the leading soil carriers in the high-risk areas of the study area, while Eutric Planosols (Ple), Calcaric Fluvisols (FLc), Eutric Fluvisols (Fle) and Humic Acrisols (Acu) do not use landslide development. The susceptibility is low.

(3) Land use: Land use classification is a unit to distinguish the spatial and geographical composition of land use, showing the way and results of land use and transformation and reflecting the form and use of land. The construction land in the study area is mainly distributed in the Yangtze River and its tributaries, and woodland accounts for more than 69% of the total area. When the land use types in the study area are Building and Unutilized land, the susceptibility is high.

5.1.5. Human Engineering Activity

(1) Road: The relevant factors of human engineering activities are negative factors for LSE. In road construction, the damage to the slope and after the completion of road construction, a series of transportation processes make the surrounding geological environment destroyed. The central and eastern parts of the road distribution are dense, the road density in the region is large, and the susceptibility is the highest when the distance from the road is between 2.0 and 5.0.

(2) Mines: The study area is rich in mineral resources, and mining activities are becoming increasingly intense, which has caused a certain degree of damage to topography. Waste rock and slag piles, tailings ponds, dumps, and transit sites will also produce environmental problems such as land occupation, and waste gas and wastewater will also be generated in production activities. The significant influence of the susceptibility is the highest nearest distance from the mine, and the susceptibility gradually decreases with the increase of distance.

According to the above statistical analysis, when the slope is 15–25°, the slope direction is West, Southwest, and North. The topography is Hill, the lithology is hard-soft-integrated, the distance from the fault is 0.5–1.0, the distance from the water system is less than 0.5, the soil types are CMd and RGc, the NDVI is 0.4–0.6, the rainfall is 1200–1400, the distance from the road is 2.0–5.0, and the distance from the mine is less than 7.0, the landslide disaster is most likely to occur in this study area.

*5.2. LSE Driving Mechanism*

5.2.1. Rainfall

Rainfall is one of the most critical factors causing slope landslides [76,77]. Since the study area is located in the subtropical climate zone with a mild and humid climate, abundant rainfall, and heavy rain or continuous rainfall, rainfall and other related functions are one of the main external forces for slope deformation and failure in this area. The rainfall driving factors in the study area can be mainly divided into early adequate rainfall and current instantaneous rainfall.

(1) Pre-effective rainfall: pre-effective rainfall is adequate before landslide formation and ultimately retained in the soil [78]. In the early stage, adequate rainfall enters the soil and remains stagnant. This rainfall process produces changes in pore water pressure and affects the stability of the slope. For soil landslides, rainfall is easy to penetrate the slope because of its loose composition. Accordingly, soil water content increases bulk density while reducing the shear strength of the sliding surface, and it is easy to induce a landslide.

(2) Instantaneous rainfall: The effect of rainfall on landslide is mainly manifested in that a large amount of rainwater infiltrates, resulting in the saturation of the soil and rock layer on the slope, and even the accumulation of water on the aquifuge below the slope, thereby increasing the weight of the landslide and reducing the shear strength of the soil and rock layer or the deformation characteristics of accumulation landslide induced by the rise of library water level [79].

5.2.2. Human Engineering Activity

The basic idea of land use prediction is to analyze and detect the driving factors of land use change between different land use types according to the characteristics of historical land use distribution and to predict the land use distribution in a certain period by using the law of land use change in the past and the demand of land use in the future. The current land use situation reflects the trend of landslides under the interference of the geological environment on human economic and social activities. The magmatic activity in the study area formed a variety of acidic to ultrabasic magmatic rocks in the pre-Sinian period and produced a series of metamorphic rock series. In addition, the tectonic activity is vigorous, and Yichang City is rich in mineral resources. Mineral development, road construction, and other engineering activities cause land occupation and topographic and geomorphological damage, which dramatically impacts the geological environment, making the LSE gradually increasing trend [80].

*5.3. Landslide Susceptibility Prediction*

It can be seen from the above that, besides the relatively static disaster-causing factors, the landslide susceptibility is also significant for its development. A landslide disaster is a dynamic evolution system. In the context of climate changes, human engineering activities, and other induced factors, the susceptibility of landslides itself will also change accordingly. Therefore, it is a new challenge to analyze the impact of changes in induced factors on landslide susceptibility and predict the trend of long-term landslide susceptibility in response to global changes. Therefore, based on the above LSE, this chapter predicts the rainfall information of the study area in 2035 and 2055 through rainfall change analysis and takes the future rainfall as a new influencing factor of landslide susceptibility analysis to predict and analyze the trend of long-term landslide susceptibility.

The original rainfall data in this paper comes from the Chinese meteorological data website. There is a high correlation between the rainfall prediction value and the historical rainfall by the atmospheric circulation model–AGCM model [81,82]. Therefore, based on the annual rainfall from 2000–to 2019 (as shown in Figure 13), we used the AGCM model to predict the annual rainfall in 2035 and 2055. The results are shown in Figure 14. The spatial distribution shows a gradual decrease in rainfall from west to east. In 2055, the overall annual rainfall increased, and the spatial distribution trend was the opposite in 2035. The rainfall showed a trend of high in the east and low in the west. Overall, in the early 21st century, the annual rainfall in the region has decreased and will increase in the middle and late stages. The trend is consistent with the impact assessment report of climate change results in the Yichang area of the Three Gorges Reservoir area.

Taking the rainfall prediction data as a new influencing factor and based on the vulnerability model established in this paper, the LSE prediction results in 2035 and 2055 can be obtained (Figure 15). The high susceptibility areas in 2035 are still concentrated in the coastal areas of the Yangtze River Basin and the main tributaries, while the low susceptibility areas are mainly distributed in high mountains and woodland areas far from rivers and low human activities. Figure 15 shows that the landslide susceptibility in 2050 has a significant change compared to 2030. The landslide susceptibility in Zigui area in the western part of the study area and Zhijiang area in the eastern part of the study area is indigenous. The landslide susceptibility in the southern part of Zigui will decrease, while the landslide susceptibility in the southeast of Zhijiang will increase. In general, the future landslide susceptibility in the study area has an overall increasing trend with the change of rainfall, while it will decrease in some areas. In the early stage, the landslide susceptibility is mainly concentrated in the western Zigui area, while in the late stage, it is transferred to the eastern Zhijiang area.

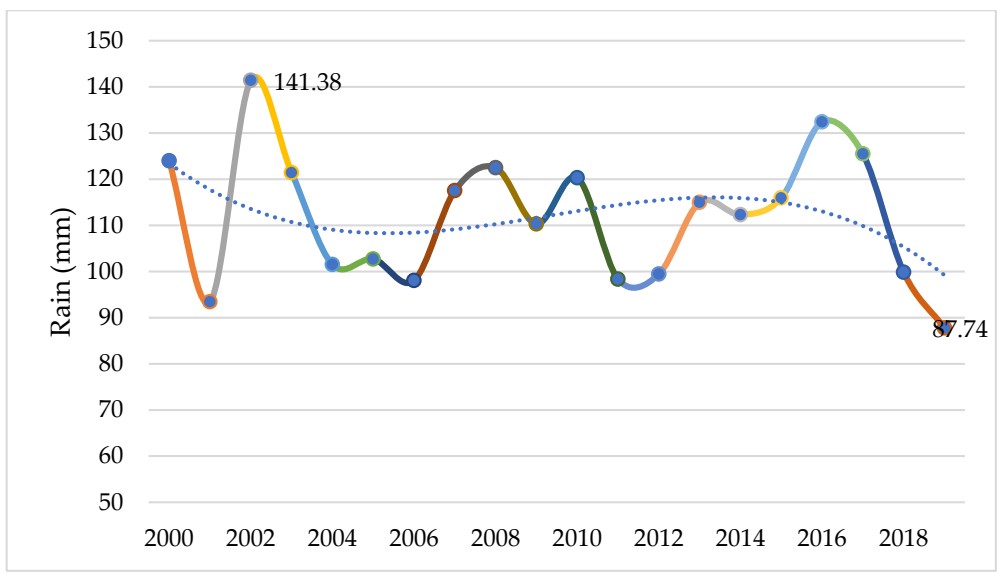

**Figure 13.** Annual rainfall change.

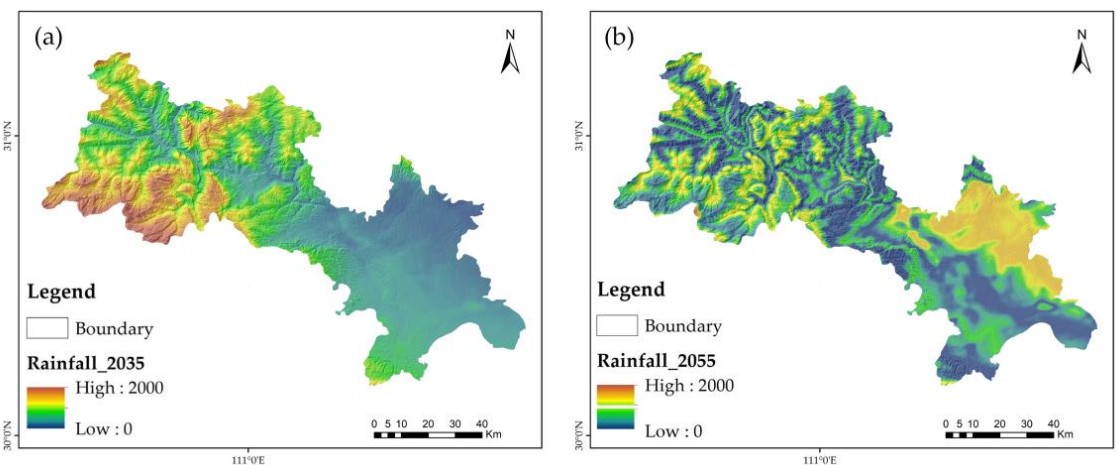

**Figure 14.** Rainfall Forecast Results. (**a**) Annual rainfall predication for 2035; (**b**) Annual rainfall predication for 2055.

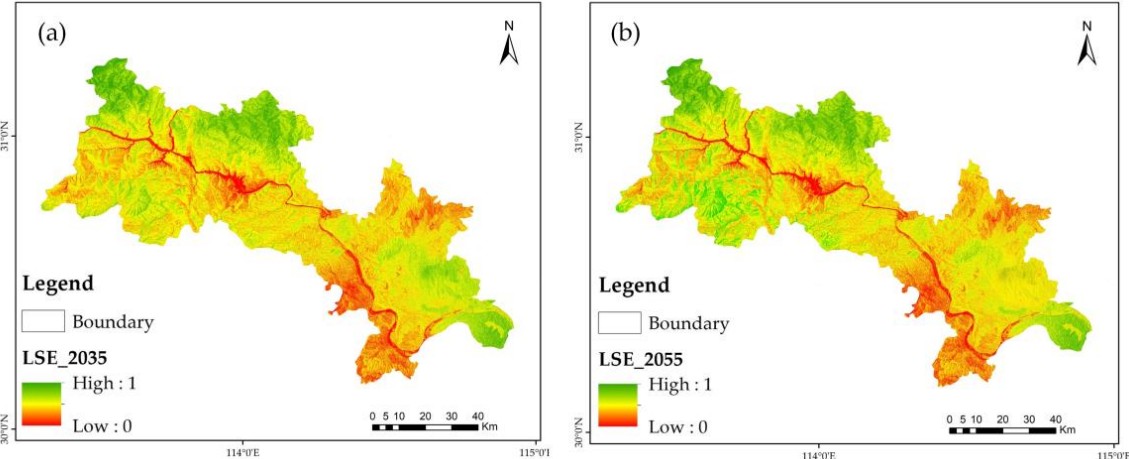

**Figure 15.** LSE Forecast Results. (**a**) LSP result for 2035; (**b**) LSP result for 2055.

The results show that the landslide disaster system is a dynamic evolution system. Under the background of climate, human engineering activities, and other environmental factors, landslide susceptibility will also change accordingly. Therefore, analyzing the impact of environmental factors on landslide susceptibility and predicting the trend of long-term landslide susceptibility can play a guiding role in dealing with the new challenges faced by landslide disaster prevention and mitigation under the conditions of global change.

## 6. Conclusions

We construct an index system to simulate the landslide susceptibility in the Yichang section based on multi-source spatio-temporal big data such as historical landslide catalog, geological data, geographical data, hydrological data, and remote sensing data of the Yangtze River Basin. For the first time, DEC and SE-CapNet deep integration networks are involved in selecting landslide samples and the training of susceptibility evaluation, revealing the primary driving mechanism of landslides in the Yangtze River Basin. At the same time, based on the constructed susceptibility model and rainfall prediction data, we carried out the mid-long term LSP and trend analysis in the study area. The conclusions and results of this paper are as follows:

(1) Based on the SE-CapNet vulnerability evaluation model after DEC non-landslide samples selection, we ensure the quality of non-landslide samples selection, retains the hierarchical relationship between factors, and automatically learns the importance of factors to enhance valuable features. Therefore, the experimental results are better than other models. The four precision index values of sensitivity, specificity, accuracy, and AUC all reach the highest values in the method comparison, which are 95.12%, 96.83%, 96.06%, and 97.3%, respectively, showing the best performance in LSE.

(2) Based on the SE-CapNet susceptibility results, the study area's hazard-causing factors and hazard-causing factors were extracted and statistically analyzed. The effects of each factor on the landslide susceptibility in the study area were evaluated, providing a reference for the subsequent LSE and variation study and providing a scientific basis for the prevention and control of landslide disasters.

(3) Based on the predicted future rainfall data as a new factor for LSE, we carried out the prediction and variation trend analysis of medium and long-term landslide susceptibility in the Yichang section of the Yangtze River Basin. The results show that with the change in rainfall in global change, the landslide susceptibility will also change accordingly. In other words, the landslide susceptibility in the study area will increase while it will decrease in some areas. In the early stage, the susceptibility is mainly concentrated in the eastern part, and in the late stage, it will be transferred to Zigui area.

Overall, this method has good performance and high precision, providing a reference for subsequent landslide susceptibility mapping, prediction and change rule research, and providing a scientific basis for landslide disaster prevention. However, many environmental and social factors affect the changes of future climate and human engineering activities. There is a certain degree of uncertainty in the future scenario predicted based on its historical change rule. Therefore, the uncertainty of predicting landslide disaster risk remains inevitable under the background of future changes.

**Author Contributions:** Data curation, C.W.; Formal analysis, R.Z.; Funding acquisition, R.Z.; Methodology, L.C.; Writing—original draft, L.C. All authors have read and agreed to the published version of the manuscript.

**Funding:** This research received no external funding.

**Data Availability Statement:** Due to the nature of this research, participants of this study did not agree for their data to be shared publicly, so supporting data is not available.

**Conflicts of Interest:** The authors declare no conflict of interest.

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
