# Peer review of "Evaluation and Prediction of Landslide Susceptibility in Yichang Section of Yangtze River Basin Based on Integrated Deep Learning Algorithm"

_remotesensing, doi:10.3390/rs14112717_

Round 1

Reviewer 1 Report

Dear authors, you have done quite interesting work on the article, which is written on the topic popular last 2 decades, and you selected a deep learning algorithm in order to assess convenient data. The results seem to be correct and your methodology can be applied to other localities.

The weakness of your article is a lot of formal issues and the quality of the presentation.

You have to resolve these issues:

  • Referencing multiple studies can be done using a hyphen in this manner [21-24].
  • Table 5 formatting needs to be improved as it is very hard to read.
  • Abbreviations that are used early on in the text need to be defined close to their first use in the text. Some abbreviations and terms are undefined.
  • Figure captions need to be more specific, especially when the figure contains multiple images. Please describe each image. Referencing the figures in the text also needs major improvement. Use their corresponding labels (a-c) when referencing them for better orientation in the text.
  • Empty spaces around brackets should be removed to make formatting better.
  • There are also missing empty spaces after some periods and before/after brackets, please run your article through a spellchecker to spot them.
  • I advise separating Figure 3 into different figures and then placing the respective figures closer to their references in the text.
  • The visibility of some figures could be another point of improvement.

Other formal issues were selected in supplemented pdf.

Author Response

Dear reviewer:

We would like to thank you and reviewers for the positive and constructive comments concerning our revised manuscript (ID: remote sensing-1720479). Those comments are all valuable and very helpful for revising and improving our paper. We have studied the comments carefully. According to the comments, we have made corrections which we hope meet with your approval. 

Reviewer 2 Report

The paper deals with the landslide susceptibility analysing the probability of landslide occurrence in a particular region considering the influence of same geological environment and trigger conditions.

The Authors show these interesting results: identification of the landslide susceptibility index system based on historical landslide inventory, geological geographic and hydrological data, remote sensing data; recognition of the main driving mechanism of landslide in the Yangtze River Basin; good accuracy of the DEC unsupervised deep clustering algorithm integrated by the SE-CapNet deep integration network.

The paper needs some clarifications.:

Figures n. 2, 7, 8, 10 and 14 must be improved (they are not very readable) 

Line 60-61: ...use.......establish (not uses and establishes)

Line 83-97: The phrase is not clear, rewrite better explaining the concept of non-landslide

Line 102-103: Explain the mining of ...negative sample....

Line 202-208: In the algorithm the distance from fault how was considered? Better explain the concept. Are some landslides seismo-induced?

Line 384: the formula is written in too large a font

Finally, I congratulate the Authors for providing valuable elements for comparing different computational approaches in a discipline as difficult to analyze as land use planning.

Author Response

(The authors gave the same response as above.)
